# Ventilator-Associated Pneumonia After Cardiac Arrest and Prevention Strategies: A Narrative Review

**DOI:** 10.3390/medicina61010078

**Published:** 2025-01-05

**Authors:** Harinivaas Shanmugavel Geetha, Yi Xiang Teo, Sharmitha Ravichandran, Amos Lal

**Affiliations:** 1Department of Internal Medicine, Saint Vincent Hospital, Worcester, MA 01608, USA; h1.shanmugavelgeetha@stvincenthospital.com (H.S.G.);; 2Division of Pulmonary and Critical Care Medicine, UMass-Baystate Hospital, Springfield, MA 01107, USA; yixiang.teo@baystatehealth.org; 3Division of Pulmonary and Critical Care Medicine Mayo Clinic, Rochester, MN 55905, USA

**Keywords:** ventilator-associated pneumonia (VAP), hospital-acquired pneumonia (HAP), cardiac arrest

## Abstract

*Background and Objectives*: Ventilator-associated pneumonia (VAP) poses a significant threat to the clinical outcomes and hospital stays of mechanically ventilated patients, particularly those recovering from cardiac arrest. Given the already elevated mortality rates in cardiac arrest cases, the addition of VAP further diminishes the chances of survival. Consequently, a paramount focus on VAP prevention becomes imperative. This review endeavors to comprehensively delve into the nuances of VAP, specifically in patients requiring mechanical ventilation in post-cardiac arrest care. The overarching objectives encompass (I) exploring the etiology, risk factors, and pathophysiology of VAP, (II) delving into available diagnostic modalities, and (III) providing insights into the management options and recent treatment guidelines. *Methods*: A literature search was conducted using PubMed, MEDLINE, and Google Scholar databases for articles about VAP and Cardiac arrest. We used the MeSH terms “VAP”, “Cardiac arrest”, “postcardiac arrest syndrome”, and “postcardiac arrest syndrome”. The clinical presentation, diagnostic, and management strategies of VAP were summarized, and all authors reviewed the selection and decided which studies to include. *Key Content and Findings*: The incidence and mortality rates of VAP exhibit significant variability, yet a recurring pattern emerges, marked by prolonged hospitalization and exacerbated clinical outcomes. This pattern is attributed to the elevated incidence of drug-resistant infections and the delayed initiation of antimicrobial treatment. This review focuses on VAP, aiming to offer valuable insights into the efficient identification and management of this fatal complication in post-cardiac arrest patients. *Conclusion*: The prognosis for survival after cardiac arrest is already challenging, and the outlook becomes even more daunting when complicated by VAP. The timely diagnosis of VAP and initiation of antibiotics pose considerable challenges, primarily due to the invasive nature of obtaining high-quality samples and the time required for speciation and identification of antimicrobial sensitivity. The controversy surrounding prophylactic antibiotics persists, but promising new strategies have been proposed; however, they are still awaiting well-designed clinical trials.

## 1. Introduction

Out-of-hospital cardiac arrest (OHCA) stands as a prominent cause of death in the United States, where a mere 8–12% manage to survive until hospital discharge. In-hospital cardiac arrest (IHCA), on the other hand, has a higher survival rate of 24% and is widely variable based on patient’s baseline characteristics and peri-postcardiac arrest care [1,2,3,4]. While early cardiopulmonary resuscitation and prompt defibrillation have proven instrumental in enhancing overall outcomes [5], the post-resuscitation phase introduces potential complications, encompassing anoxic brain injury, myocardial dysfunction, and systemic ischemia. Furthermore, infections such as ventilator-associated pneumonia (VAP) can significantly impact highly vulnerable patients following the restoration of spontaneous circulation (ROSC). VAP is characterized by the onset of pneumonia within 48–72 h after endotracheal intubation, with the mean duration from intubation to VAP development of 3.3 days, and the risk declines daily. Numerous studies indicate that VAP is a prevalent form of hospital-acquired pneumonia [6,7,8].

Furthermore, VAP has been observed to frequently lead to prolonged mechanical ventilation and extended hospitalization, with a mortality rate ranging from 24% to 50% [8,9,10]. It is essential to note that this mortality rate is variable and heavily dependent on comorbidities. This review delves into crucial aspects of VAP, specifically focusing on unraveling insights into its development among cardiac arrest patients. Our objective is to enhance comprehension of the disease, paving the way for identifying potential treatment and prevention strategies through thoroughly examining these key facets.

## 2. Methods

This review summarizes the clinical presentation and management of patients with cardiac arrest complicated by VAP. We searched PubMed, MEDLINE, and Google Scholar databases for articles on VAP in Cardiac arrest. We used the MeSH terms “VAP”, “Cardiac arrest”, “postcardiac arrest syndrome”, and “postcardiac arrest syndrome”. Studies reporting patient demographics, clinical presentations, and management of VAP in cardiac arrest patients were included in this review. Incorporated publications included clinical trials, cohort studies, and case-control studies published before December 2023. In addition, the bibliography of selected articles was examined for further studies. We included only studies published in English and excluded articles that included opinions, letters, abstracts, and preprints yet to undergo peer review.

## 3. Definition

The categorization of pneumonia hinges predominantly on the site of acquisition, delineating between community-acquired pneumonia (CAP) and hospital-acquired pneumonia (HAP) (Table 1). CAP encompasses infections contracted beyond healthcare settings or within the initial 48 h of hospital admission. *Streptococcus pneumoniae* takes precedence as the most prevalent bacterial cause, closely followed by *Haemophilus influenza* and *Moraxella catarrhalis* [11,12].

In contrast, HAP, which manifests 48 h after admission, includes subtypes known as ventilator-associated pneumonia (VAP) and non-ventilator-associated HAP (nvHAP).

VAP is characterized by the development of pneumonia in intubated patients for more than 48 h and within 48 h of extubation.nvHAP, on the other hand, specifically pertains to instances of pneumonia developing in hospitalized patients who are neither on mechanical ventilation nor have undergone extubation within 48 h before the onset of pneumonia.

Pathogens are similar among VAP and nvHAP, with the most common organisms being *Staphylococcus aureus* (including *methicillin-resistant S. Aureus*) and *Pseudomonas aeruginosa* [13,14].

Notably, HCAP was introduced in 2005 by the American Thoracic Society and Infectious Disease Society of America (ATS/IDSA) to identify patients at risk for infection with multidrug-resistant (MDR) pathogens due to their healthcare exposures. The United States centers for Disease Control and Prevention (CDC) created the ventilator associated events (VAE) definition to replace the traditional definitions for ventilatory associated pneumonia (VAP). Although a number of studies have been performed to assess the overlap between VAE and VAP, there appears to be a mismatch. Hence, VAE has been defined to be a new entity as a safety surveillance tool to encompass a broad range of potential nosocomial complications.

However, the 2016 and 2019 ATS/IDSA guidelines and the 2017 European and Latin American guidelines for hospital-acquired pneumonia (HAP) no longer recognized HCAP as a separate category (Table 1). This change was based on the recognition that the HCAP designation did not consistently identify patients at risk for resistant pathogens, and the evidence supporting its use was considered of low quality [12,15,16,17,18].

## 4. Epidemiology

More than 500,000 adults in the United States experience cardiac arrest each year, encompassing both cases of OHCA and IHCA [19]. OHCA is frequently attributed to cardiac diseases, notably coronary artery disease (CAD), and structural heart disease.

Conversely, IHCA often arises from the progression of respiratory distress [20], with the etiology varying based on the type and location [21]

Mild therapeutic hypothermia (MTH), where comatose patients were actively cooled with a surface or intravascular temperature-management device to a target temperature of 33 °C, has been a common practice in postcardiac arrest care. This was aimed at reducing the inflammatory response following ROSC. However, in 2021, Dankiewicz et al. found that MTH does not lead to significant clinical benefits but increases the risk of adverse events [22]. Hasslacher et al. subsequently found that patients treated with MTH had an odds ratio of 2.67 for developing VAP [23].

The crude mortality rate for VAP spans a substantial range, fluctuating between 10% and 55% [16,24,25,26]. The timing of pneumonia onset also emerges as a significant determinant, with late-onset VAP being particularly concerning due to its likely association with multidrug-resistant pathogens [27].

Diagnosis of VAP in patients with cardiac arrest is complex due to the nonspecific nature of its clinical presentation, which often overlaps with other conditions commonly seen in critically ill patients. Fever, leukocytosis, hypoxemia, and radiographic infiltrates, hallmarks of VAP, can also be caused by noninfectious processes such as acute respiratory distress syndrome (ARDS), atelectasis, pulmonary thromboembolism, pulmonary hemorrhage, or drug-induced pulmonary injury. Similarly, systemic conditions like congestive heart failure or hemodynamic instability following cardiac arrest can mimic VAP symptoms, further complicating diagnosis.

## 5. Etiology

Introducing a foreign body, such as the endotracheal tube, can be a nidus for infective pathogens. A study by Rello et al. reported that pneumonia developed in 24% of their population, and all were undergoing mechanical ventilation [28]. The causative agents of VAP exhibit variability depending on patient demographics, duration of hospitalization, and the diagnostic approach. Despite this diversity, a consistent pattern emerges across numerous studies, with *Pseudomonas aeruginosa* (10–55%) and *Staphylococcus aureus* (7–44%) being the predominant pathogens implicated in VAP cases and less commonly by *streptococcus pneumoniae*, *enterococcus faecalis*, and fungal [6,9,25,29,30,31,32,33,34,35] (Table 2). Notably, a recurrent emphasis on the high incidence of polymicrobial infection has been observed. In a study by Fagon et al., invasive diagnostic methods, such as bronchoscopic protected sampling, yielded a higher percentage of monobacterial isolates30.

*Pseudomonas aeruginosa* poses a formidable challenge in the treatment of HAP and VAP, given the escalating prevalence of multidrug-resistant strains observed over the past decade. The surge in multidrug resistance (MDR) has been accompanied by a notable twofold increase in the risk of mortality associated with MDR *P. aeruginosa* [36,37,38]. A key risk factor identified for Pseudomonas VAP is recent antimicrobial use within the past 90 days. Moreover, individuals with prior *P. aeruginosa* colonization, such as those with chronic obstructive lung disease and cystic fibrosis, face an eightfold higher odds of developing *P. aeruginosa* VAP compared to patients not colonized by these bacteria. This colonization is further associated with local resistance patterns [6,39]. Additionally, the administration of inadequate initial antibiotic therapy in suspected VAP cases has been linked to heightened mortality rates and prolonged stays in the ICU [37].

While the prevalence of *Pseudomonas aeruginosa* in VAP remains relatively consistent worldwide, certain regions, including the Middle East, India, East Asia, and South America, exhibit a distinctive prevalence of *Acinetobacter baumannii* as a prominent infectious etiology in VAP [25,33,34,40,41]. Acinetobacter species are opportunistic gram-negative bacilli commonly found in warm and moist environments, and their increasing prevalence in intensive care units globally has become a significant concern due to widespread antimicrobial resistance.

*Staphylococcus aureus* is commonly present in the orolaryngeal and tracheobronchial flora. The risk of nosocomial infection is heightened with endotracheal intubation, as this procedure disrupts the immune defenses provided by the mucosa. *S. aureus* is notorious for causing severe diseases attributed to the production of virulence factors. These factors facilitate the initiation of colonization by enabling adherence to tissues, subsequent damage to these tissues, and the potential dissemination to other organs [42,43].

Although invasive fungal VAP was reported, it appears to have a lower prevalence in causing VAP. However, Huang et al. have highlighted a noteworthy concern regarding the association between antimicrobial use and mechanical ventilation, which elevates the risk of Candida colonization. This, in turn, significantly increases the likelihood of carbapenem-resistant infections, particularly involving *Pseudomonas aeruginosa* (21%), *Klebsiella pneumonia* (14%), and *Acinetobacter baumannii* (22.5%) [33].

## 6. Pathophysiology

The Pathophysiology underlying the development of VAP in patients who underwent cardiac arrest is actively being studied and is postulated to be due to multifactorial etiology. The trachea is normally colonized by endogenous flora, but the lower respiratory tree is sterile. VAP develops with a shift in balance when microbial pathogens enter the lower respiratory tree, followed by colonization, in conjunction with disruption in host response with impaired neutrophil function, especially in critically ill patients, increasing the risk of acquiring nosocomial infections. Several host- and treatment-related colonization factors, such as the severity of the patient’s underlying disease, prior surgery, exposure to antibiotics and other medications, and exposure to invasive respiratory devices and equipment, are important in the pathogenesis of VAP [44,45].

Patients are treated with therapeutic hypothermia post cardiac arrest to impair the general inflammatory response, thus exerting protective effects on the brain, But this affects the ability of the immune system to mount an appropriate response and may suppress leukocyte migration and phagocytosis, thus increasing the risk of any infectious complications. Hasslacher et al. noted that VAP that developed in patients after treatment with mild therapeutic hypothermia was accompanied by more ventilator-dependent days, prolonged antibiotic treatment, and ICU-LOS [23].

The primary entry route of infection in endemic VAP is the lungs, through aspiration of pathogenic organisms that have colonized the oropharyngeal tract (or, to a lesser extent, the gastrointestinal tract) [46].

Epidemic VAP occurs from microbes acquired from the hospital environment. Safdar N et al. noted that as many as 75 percent of severely ill patients will be colonized within 48 h [47]. An additional mechanism of inoculation in mechanically ventilated patients is direct contact with environmental reservoirs, including respiratory devices and contaminated water reservoirs [6,48]. Other sources of infection including endotracheal cuff biofilm that embolizes to the distal airways, inhalation from contaminated aerosols, hematogenous spread, or translocation from the gastrointestinal tract lumen are rare.

## 7. Clinical Features

A comprehensive history and physical examination are warranted to help diagnose VAP, assess the severity, rule out other causes of the symptoms, and reveal the potential etiology of the disease.

The 2016 Infectious Diseases Society of America/American Thoracic Society (IDSA/ATS) guidelines for the management of HAP and VAP recommend a clinical diagnosis based upon a new lung infiltrate plus clinical evidence that the infiltrate is of infectious origin, which includes the new onset of fever, purulent sputum, leukocytosis, and decline in oxygenation, occurring gradually or as a sudden onset, more than 48 h after intubation [18].

These signs and symptoms are nonspecific; individually, they do not have a high sensitivity or specificity for diagnosing VAP. Patients may also have complications of VAP, including hypoxia, reduced tidal volume, increased inspiratory pressures, hemoptysis, bronchospasm, sepsis and septic shock, and multiorgan failure.

## 8. Diagnosis & Testing Strategies

The diagnosis of VAP starts with clinical suspicion of an infection, followed by imaging and microbiological confirmation.

### 8.1. Clinical Strategy

A suspicion of VAP occurs with at least two of the clinical features defined by the 2016 IDSA/ATS guidelines, starting at least 48 h after intubation. In the presence of ARDS, even one of the following should lead to a suspicion of VAP [18].

Radiographic evidence of new or progressive infiltrates is necessary to diagnose VAP and is the second step in the diagnosis after a clinical suspicion.

Chest X-rays can be easy to obtain and less expensive but are neither sensitive nor specific for VAP. CT chest without contrast can be more sensitive and useful when a chest x-ray is normal, but a high clinical suspicion exists for VAP. It can also be useful for finding the target lobe for sampling and better looking for progression from prior pneumonia. Lung ultrasound can be a less expensive diagnostic aid, but data on its sensitivity and specificity still need to be provided.

Other tests with limited value in diagnosing VAP include procalcitonin (PCT), CRP, and soluble triggering receptors on myeloid cells (sTREM 1). The 2016 IDSA/ATS guidelines do not recommend using PCT along with clinical criteria for routine diagnosis but suggest using the combination to discontinue antibiotic therapy. The rationale is that no evidence suggests improved outcomes in using PCT for diagnosis, and the sensitivity and specificity failed to meet the panel’s prespecified thresholds. However, for discontinuation, the panel felt that the benefits of decreased antibiotic exposure outweighed the costs, burdens, and uncertain results associated with PCT testing [18] The 2017 International guidelines recommended against the use of PCT with clinical criteria to consider stopping antibiotics, especially if the duration is only 7–8 days. They also recommended against using PCT along with bedside clinical assessment in patients receiving antibiotic therapy to predict adverse outcomes and clinical response at 72–96 h since the panel considered the burden of the potentially high costs outweighed the benefits of their limited prognostic capacity [16].

Advocates of the clinical strategy recommend initiating empiric antibiotic therapy once clinical suspicion with new or progressive infiltrates are identified. A decision about antibiotic discontinuation can be made on day 2 or 3, assessing clinical response and when semiquantitative cultures of lower respiratory tract samples are available [15].

### 8.2. Microbiological Strategy

Once VAP is suspected with clinical findings and imaging, a positive microbiological culture is essential to confirm the diagnosis of VAP, and a sample must be collected as soon as possible, preferably before initiating antibiotic therapy. This is because a clinical suspicion alone can lead to overtreatment, and culture data can be used to de-escalate or narrow antibiotics during the course.

A sterile culture of respiratory secretions in the absence of a new antibiotic in the past 72 h virtually rules out the presence of bacterial pneumonia. However, a viral or legionella infection is still possible [49]. Samples can be obtained by non-invasive (Endotracheal tube aspiration) or invasive [e.g., bronchoalveolar lavage (BAL), mini-BAL, protected specimen brush (PSB)] meth ds. Cultures can be quantitative (pathogens reported in the number of colony-forming units), semiquantitative (reported as heavy, moderate, light, or no growth), or qualitative (reports presence or absence of pathogens).

Invasive methods of collecting samples, though they can be time-consuming and require qualified personnel, increase diagnostic accuracy by differentiating from colonization since the samples are collected from the lower respiratory tract.

Quantitative cultures, with results above the diagnostic threshold for VAP, also help differentiate VAP from colonization, promoting good antibiotic stewardship compared to non-invasive sampling (i.e., endotracheal aspiration) and nonquantitative cultures. Expenses and availability, however, limit the use of quantitative cultures. Though invasive sampling and quantitative cultures have a higher specificity, Berton et al. l observed that there is no evidence that their use results in reduced mortality, reduced time in ICU and on mechanical ventilation, or higher rates of antibiotic change when compared to non-invasive sampling and qualitative cultures in patients with VAP [50].

The 2017 International ERS/ESICM/ESCMID/ALAT guidelines for diagnosing and managing VAP differ from the 2016 clinical practice guidelines by IDSA/ATS (Table 3).

Blood cultures are to be obtained in all patients with AP. The sensitivity of blood cultures for the etiologic diagnosis of VAP is less than 25%. A positive culture can indicate an extrapulmonary source in a larger percentage than a pulmonary source, even if VAP is also present [51].

### 8.3. Molecular Diagnosis

One of the major drawbacks of the microbiological culture assays was the yield and the delay in the time of diagnosis. Hence, molecular diagnostic methodologies such as Polymerase chain reaction (PCR) can be utilized to rapidly diagnose patients and ascertain the diagnosis [52].

## 9. CPIS

The Clinical Pulmonary Infection Score (CPIS) is a semi-objective tool designed by Pugin et al., to increase the specificity of the diagnosis and assist clinicians who historically have been relying on clinical criteria in deciding to initiate antibiotic therapy in AP. It combines clinical, radiographic, physiological, and microbiological data into a numerical record. A score greater than six was consistent with the diagnosis of VAP [53]. However; a subsequent study showed that the CPIS could diagnose VAP with a sensitivity and specificity of only 65% and 64%, respectively [54]. The 2016 IDSA/ATS guidelines do not suggest the use of CPIS along with clinical criteria in deciding to initiate or discontinue antibiotic therapy in AP. The false-negative and false-positive rates of the CPIS are 35% and 36%, respectively, assuming a prevalence of VAP of 8%. The panel agreed that the frequency of such undesirable consequences due to misleading CPIS results was unacceptable and, therefore, recommended not using CPIS to guide antibiotic therapy [18].

The advantages and disadvantages of certain clinical and microbiological strategies in the diagnosis of VAP are outlined below in Table 4.

Although there has been a recent increase in the utilization of different diagnostic strategies and resources to identify VAP earlier and in turn help reduce the significant healthcare burden associated with increased hospital and ICU stay, several of these strategies have a high cost associated with them. This reduces the availability and accessibility of these resource in developing countries and resource limited economies.

## 10. Management

### 10.1. Timing of Initiating Antibiotics

Once a clinical diagnosis of VAP has been made, prompt initiation of antibiotic therapy is emphasized. A delay in initiating appropriate antibiotic therapy is associated with increased mortality [52,55,56]. Obtaining a sample for microbiological diagnosis before initiating antibiotics is recommended. Occasionally, it might not be feasible to obtain an invasive sample like BAL before antibiotic therapy, especially if the patient presents with complications like septic shock and respiratory deterioration requiring initiation of antibiotics as soon as possible. A mini-BAL or an endotracheal aspirate can be considered in such situations.

### 10.2. Choosing an Empiric Therapy

Tailoring antibiotic therapy to the culture and drug susceptibility data would be ideal for decreasing the development of resistant pathogens. This is limited by the delay in obtaining samples, especially invasive ones, and a delay in reporting microbiological data. It takes up to 24–36 h to obtain a positive culture and up to 48–72 h to obtain an antimicrobial sensitivity report [57]. Microscopic exam and gram staining of the sample can shed some light on the possible pathogen and help guide the antibiotic choice before culture results. Studies have shown that less use of antibiotics is associated with no adverse outcomes and improved mortality [30,58]. However, some investigators agree not to withhold antibiotics based on these initial ancillary tests in patients who have a high pretest probability of pneumonia and are clinically unstable [30,59].

When choosing the right empiric therapy, the aim is to avoid unnecessary antibiotics and provide the initial appropriate therapy. Since culture data is not readily available, the selection of the appropriate initial empiric antibiotic therapy is determined by the risk factors for multi-drug resistant (MDR) pathogens and the local prevalence of organisms and drug susceptibility [15].

Below are the recommendations for choosing empiric therapy from the 2016 Clinical Practice Guidelines by the Infectious Diseases Society of America and the American Thoracic Society and the 2017 International ERS/ESICM/ESCMID/ALAT guidelines (Table 5).

In addition to the above combination of empiric therapy for high-risk organisms, the below should be considered if specific organisms are suspicious. If an ESBL+ strain, such as *K. pneumoniae* or an *Acinetobacter* species, is suspected, third-generation cephalosporins must be avoided, and the carbapenem must be used. For carbapenem-resistant Acinetobacter species, sulbactam, colistin, and polymyxins are active agents [15]. Aerosolized antibiotics have not been proven to have value in the therapy of VAP, but they may be considered as adjunctive therapy in patients with MDR gram-negatives who are not responding to systemic therapy [15,60].

### 10.3. Combination vs. Monotherapy

Combination regimens have been used for reasons like achieving synergy in the therapy of *P. aeruginosa* and preventing the emergence of resistance during therapy. Since they are mostly MDR pathogens, providing a broad spectrum regimen with combination therapy from two different classes that include at least one drug that is active against the MDR pathogens, when suspected, would decrease the chances of inappropriate treatment [15].

### 10.4. Modification of Therapy

Clinical improvement usually occurs in 48–72 h, and thus, assessing response to empiric therapy on days 2–3 is ideal unless there is a rapid clinical decline. Patients with no clinical improvement should be evaluated for alternate diagnoses, organisms not covered by the initial empiric therapy, drug resistance, side effects, complications of pneumonia, and other sources of infection. In patients who respond, de-escalating antibiotics and narrowing therapy based on culture and susceptibility data is recommended.

### 10.5. Duration of Therapy

Traditionally, the duration of antibiotic therapy was 14 to 21 days. Literature shows no difference between short courses (7–8 days) and longer courses (10–15 days) in mortality rates, recurrent pneumonia, treatment failure, hospital length of stay, or duration of mechanical ventilation. However, for *P. aeruginosa* and *Acinetobacter*, the relapse rate was higher with a shorter duration of treatment. The 2016 IDSA/ATS guidelines recommend a 7-day course of antibiotic therapy rather than a longer course [18]. The 2017 European guidelines also recommend a 7–8 day course for patients without complications like immunocompromised, empyema, lung abscess, cavitation, or necrotizing pneumonia and with good clinical response [16]. Singh et al. used a modification of the CPIS score to identify low-risk patients (with a CPIS score of 6 or less) to consider discontinuing antibiotics on da 3. Patients showed better clinical outcomes with fewer subsequent superinfections [63].

In summary, initiating empiric therapy based on risk factors for MDR pathogens and local prevalence and susceptibility data as soon as clinical suspicion of VAP arises is recommended to decrease mortality. If *P. aeruginosa* is suspected and there is a risk for high-risk disease, combination therapy including drugs from two different classes is recommended by the 2016 Clinical practice guidelines by IDSA/ATS [18]. Clinical response is assessed on days 2–3 of therapy, and the de-escalation of antibiotics should be considered based on culture data to prevent the emergence of resistant pathogens.

## 11. Complications/Outcomes

Complications arising from mechanical ventilation and critical illness after cardiac arrest have significantly changed over the past few years. As complications such as cardiovascular disease, renal failure, and sepsis continue to grow, other respiratory complications such as pneumonia and barotrauma have reduced, as demonstrated in the study by Esteban et al. [64]. Although the incidence of pneumonia is reducing, recent studies demonstrate that ventilator-associated pneumonia often leads to a host of other complications that consequently result in multiorgan failure and in turn higher mortality. Septic shock, acute respiratory distress syndrome, atelectasis, and infection with multidrug-resistant organisms were seen more commonly in patients with ventilator-associated pneumonia [65]. One possible explanation for VAP leading to the incidence of other complications is the greater length of stay associated with VAP, thus resulting in a higher rate of complications. Studies demonstrate that patients with VAP had increased duration of mechanical ventilation, Length of ICU stay, and length of hospitalization [66]. The mortality rate associated with ventilator-associated pneumonia is between 9 and 27%, and infection with organisms such as Pseudomonas and Acinetobacter increases the mortality rate [65,67]. Another important factor influencing the mortality rate is the day of antibiotic administration. Delay in starting the appropriate antibiotics progressively increases the mortality rate associated with ventilator-associated pneumonia [8].

In addition to the clinical complications associated with VAP, there appears to be a huge economic burden posed by the disease. The retrospective study done by Rello et al. encompassing 9080 study participants and analyzing the resource utilization and hospital charges associated with VAP demonstrates that VAP is associated with an increase of >USD 40,000 in mean hospital charges per patient when compared to patients who did not develop VAP [6]. Other similar studies such as that by Safdar et al., and Kollef et al. agree with the findings of increased hospital charges associated with VAP [66,68].

## 12. Risk Factors

Considering the increased burden of complications associated with VAP, it becomes pivotal to understand the risk factors associated with VAP, which will help us institute appropriate prevention strategies (Figure 1).

### 12.1. Age

Initial trials postulated an increased VAP incidence in patients with advanced age, attributing a 1.15 to 1.5-fold risk with an increase in age. They attributed the increased risk to the physiological aging process and concomitant pharyngeal muscle weakness, loss of protective cough reflex, and weakened immune system [69,70,71]. However, further studies, such as the multicenter cohort by Blot et al., demonstrated no significant association between advanced age and increased risk for AP [72]. Zubair et al. demonstrated increased mortality in older patients with VAP, suggesting that advancing age could be independently related to VAP incidence and mortality [73]. The literature evidence suggests conflicting data regarding the association of age as an independent risk factor and VAP incidence. However, with the increased number of medical comorbidities associated with advancing age, there appears to be an increased predisposition to VAP risk in older patients.

### 12.2. Male Sex

The other patient characteristic associated with VAP risk is the Male sex. Unlike Age, most studies suggest an increased VAP risk in men when compared to women [74]. Tejerina et al. demonstrated that men had a 1.3-fold increased risk of developing VAP compared to women, and this was supported by multivariate analysis as well [75,76].

### 12.3. Prolonged Mechanical Ventilation

Prolonged mechanical ventilation has been attributed to increased physical deconditioning, resulting in loss of protective reflexes, increased risk of aspiration of gastric contents, and greater VAP risk. Although this is supported by literature, other studies demonstrate prolonged mechanical ventilation risk due to VAP, raising the question of whether prolonged mechanical ventilation leads to VAP or whether VAP leads to prolonged mechanical ventilation [70]. Currently, we do not have any definitive literature supporting one hypothesis over the other.

### 12.4. Altered Mental Status

Patients with a Glasgow Coma scale of less than eight were found to have increased VAP-associated mortality [71]. Jovanovic et al. reported pre-admission coma as an independent risk factor for VAP [77]. In patients with altered mental status, loss of protective respiratory reflexes like swallowing and coughing can lead to aspiration of secretions resulting in VAP.

### 12.5. Chronic Obstructive Pulmonary Disease (COPD)

Lung pathologies such as Chronic obstructive pulmonary disease(COPD) are commonly associated with increased VAP risk, with studies suggesting 2.35 times higher risk [75]. Since patients with COPD tend to have prolonged mechanical ventilation, this, in turn, poses a risk factor for AP. Although the VAP risk of other lung diseases hasn’t been adequately studied, all diseases that lead to prolonged mechanical ventilation naturally increase VAP risk.

### 12.6. Smoking History

Amongst the relevant past medical history, smoking has been found to have a 4.37 increased fold of VAP incidence [69]. This is attributed to the impaired ciliary function and host defense mechanisms commonly associated with smoking.

### 12.7. Other Diseases

Studies analyzing the epidemiology and risk factors associated with VAP showed that other diseases such as Coronary artery disease, Diabetes mellitus, Hashimoto’s thyroiditis, chronic respiratory failure, and spinal cord injury have been associated with increased VAP risk [78,79,80].

## 13. Procedures

Invasive medical procedures are often associated with increased VAP risk. Procedures such as re-intubation, tracheostomy, and bronchoscopy are associated with increased VAP risk [81,82]. The impairment of the normal epiglottic barrier increases the risk of aspiration and subsequent VAP incidence. Bronchoalveolar lavage and indwelling gastric tubes have also been associated with increased risk [7].

## 14. Aspiration Prevention Strategies

### 14.1. Non-Invasive Positive Pressure Ventilation

With intubation and re-intubation associated with increased VAP risk, the utility of non-invasive positive pressure ventilation is availed in managing respiratory diseases as a preventive strategy to prevent VAP incidence. Nava et al. demonstrated that noninvasive pressure support during weaning decreases the incidence of pneumonia and also improve 60 day mortality. Antonelli et demonstrated that patients on noninvasive ventilation improved gas exchange as well as had fewer complications and shorter ICU stays [83,84,85].

### 14.2. Head of Bed Elevation

In addition to aspiration of secretions, aspiration of gastric secretions often leads to increased VAP incidence. Supine positioning in addition to long duration of mechanical ventilation and decreased consciousness lead to increased risk of aspirations. Drakulovic et al. first demonstrated that semirecumbent body position reduced the frequency and risk of VAP [86]. This study was based on the fact that several studies showed increased aspiration risk in supine patients. However semirecumbent positioning does not protect completely from gastroesophageal reflux [87,88].

Preventive strategies are highlighted in Figure 2.

### 14.3. Sedation Holiday

Protective airway reflexes are a key component in preventing VAP; any factor affecting them would increase VAP risk. One such factor is the sedation administered for intubated patients. Sedation holidays and daily weaning are key strategies for reducing the VAP risk posed by sedation. This has been postulated since studies suggest decreased mechanical ventilation duration in patients regularly on sedation holiday and mechanical ventilation being classified as a risk factor for VAP. The Study by Brook et al. showed that protocol directed sedation reduced the duration of mechanical ventilation as well as ICU and hospital stay. They attributed it to reduction in complications such as VAP incidence [89,90]. However, we do not have direct evidence supporting the beneficial effects of sedation holiday in reducing VAP risk.

### 14.4. Subglottic Suctioning

Endotracheal intubation has often been associated with increased VAP risk due to the disruption of the epiglottic barrier, aspiration of secretion and gastric contents, and formation of biofilm. Subglottic suctioning was postulated to decrease the incidence of VAP by constantly removing the pool of secretions that may be aspirated. Although initially there was no clear data and the findings were controversial, recent meta-analyses prove that subglottic suctioning leads to decreased VAP incidence and mechanical ventilation duration [91,92].

### 14.5. Antimicrobial-Coated Endotracheal Tubes

Since Biofilm production in the endotracheal tube is also a common mechanism postulated to cause VAP, the use of antimicrobial coating on ET tube has been proposed to decrease VAP risk [93]. The North American Silver-Coated Endotracheal tube (NASCENT) study looked into the utility of silver-coated endotracheal tubes in decreasing VAP risk as silver has been shown to have broad-spectrum antimicrobial activity blocking biofilm formation [94,95]. The study showed a decreased incidence of VAP compared to conventional tubes however these tubes were relatively expensive. Although other coating materials, such as chlorhexidine and sulphadiazine, have been proposed, we need studies to confirm their utility in real world [96].

### 14.6. Decolonization

Selective decontamination of the digestive tract and oropharynx with broad-spectrum antibiotics was proposed to reduce VAP risk. This was demonstrated by Smet et al. in a prospective study done in the Netherlands that showed a reduction in the mortality rate in patients who underwent selective decontamination of the digestive tract and oropharynx [97]. However, another study by Oostdijk et al. showed that routine decontamination resulted in an increased prevalence of antibiotic resistance questioning the long-term utility of routine decontamination.

Selective oral decontamination with agents such as chlorhexidine or povidone-iodine has been demonstrated to reduce VAP risk, but their impact on antibiotic resistance is yet to be studied [98,99].

### 14.7. Probiotics

The advent of the Gut-lung axis terminology has sparked interest in studying probiotics to prevent severe respiratory infections. Their utility in VAP has also been studied, and the clinical trial by Morrow et al. demonstrates the efficacy of probiotics in preventing VAP incidence in high-risk ICU patients. Their study showed that patients treated with *Lactobacillus rhamnosus* were less likely to to develop microbiologically confirmed VAP [100].

### 14.8. Early Mobilization

Early mobilization has been attributed to improve physical conditioning, and this, in turn, reduces weakness caused by critical care illnesses. The Meta-analysis by Zang et al. demonstrated a significant reduction in VAP risk when early mobilization was employed. The study by Wang et al. showed that early mobilization reduced the incidence of intensive care unit-acquired weakness and complications such as VAP, deep vein thrombosis, pressure sores and also reduced the duration of mechanical ventilation, length of ICU stay and hospital stay [101,102].

### 14.9. Early Enteral Nutrition

The espen guidelines recommend early enteral nutrition over parenteral nutrition to reduce the vap risk since it would stimulate gut peristalsis, thereby decreasing the risk of aspiration. special attention was recommended to the provision of glutamine and omega-3 fatty acids as well as the care of patients such as those with dysphagia, frail patients, multiple trauma patients, abdominal surgery, sepsis and obesity [103].

### 14.10. Post Pyloric Feeding

Post-pyloric feeding has been postulated to reduce the risk of aspiration by utilizing the strong duodenal sphincter mechanism, and this was supported by a meta-analysis by Liu et al. 41 studies were included across 10 countries involving 3248 patients and the results showed that post-pyloric feeding was associated with lower rate of pulmonary aspiration, gastric reflux, gastrointestinal complications and VAP [104].

### 14.11. Low Tidal Volume Ventilation

Low tidal volume ventilation has been shown to reduce the risk of pressure and volume injury to the alveoli, thereby reducing the rate of ventilator-associated events [105]. However, recent studies also suggest that low-volume ventilation reduces the risk of VAP. Neto et al. studied patients on low tidal volume ventilation compared to those not on it and found a lower number of ICU free days, hospital free days and a greater mortality and complication rate amongst patients not on low tidal volume ventilation [106,107].

### 14.12. Restrictive Transfusion Threshold

Blood transfusions have been shown to lower immunity and increase the risk of healthcare-associated infections. Rohde et al. demonstrated that utilizing a restrictive transfusion strategy reduces the risk of infection [108].

### 14.13. Use of Prophylactic Antibiotics

Despite significant advances in the understanding of the incidence of ventilator-associated pneumonia in patients who suffered a cardiac arrest and more than 30 years of research, the use of prophylactic antibiotics in cardiac arrest patients to prevent ventilator-associated pneumonia has been a subject of debate. While it has been argued that the use of prophylactic antibiotics has been associated with the merit of VAP prevention, there is a significant argument on the harm caused by them by leading to the resistance of multi-drug resistant organisms and events such as nephrotoxicity [109]. The recent meta-analyses by Zha et al. along with previous studies suggest that the use of prophylactic antibiotics has been evident in reducing the risk of VAP but not in reducing the mortality rate. This raises questions since the development of VAP is an independent risk factor for increased mortality. These meta-analyses also state that most studies included were of low evidence and the need for well-designed randomized studies with large sample sizes [109,110,111]. Mirtalaei et al. demonstrate the utility of prophylactic antibiotics extends only to the prevention of early-onset VAP and not to late-onset VAP indicating that there might be more to the pathogenesis of VAP that would help us better understand its incidence [112]. A recent multi-center trial in France demonstrated the utility of prophylactic ceftriaxone administration in the prevention of VAP [113].

### 14.14. Nebulized Antibiotics

One of the significant issues attributed to the prophylactic use of antibiotics is the rise of systemic complications such as nephrotoxicity, and nebulized antibiotics have been proposed to circumvent this complication by minimizing systemic exposure [114]. Nebulized antibiotics have been postulated to pose several advantages over systemic antibiotics such as,

By achieving greater respiratory levels of drug concentration, they exceed the Minimum inhibitory concentration (MIC) of bacteria, thereby decreasing resistance emergence [115]Reducing systemic complications by parenteral antibiotics such as nephrotoxicity [116]Direct action on antibiotic film formation on endotracheal tubes [117]

Despite the proposed benefits, initial studies did not reveal any utility of nebulized antibiotics in VAP prevention and treatment [118,119]. The meta-analysis by Xu et al. showed that definitive conclusions could not be inferred due to the low level of evidence and the need for large randomized studies [120]. This is demonstrated in the ATS guidelines that also state that aerosolized antibiotics have not been proven to be useful in the treatment of VAP but may be considered as an adjunct in patients with multidrug-resistant gram-negative bacterial infection [15]. However, recent well-randomized control trials such as that by Ehrmann et al. show that a three-day course of inhaled amikacin reduced the burden of AP [121]. We also have emerging studies that prove similar benefits [110]. Further well-randomized studies and meta-analyses are essential to understand better the utility of nebulized antibiotics in VAP treatment and prevention.

Several of the VAP prevention strategies encompass the utilization of antibiotics in systemic and topical modalities to prevent the incidence of VAP. Although several of these strategies have been proven to be beneficial, there is always greater concerns of antibiotic resistance after long term use of antibiotic resistance. We would require studies of longer duration to assess the impact of these strategies in the long term antibiotic stewardship.

The other limiting factor that is associated with the utilization of these management strategies is availability of health care funding to invest in one or several of these modalities. The health expenditures per capita vary largely between developing and developed countries and hence allocation of funds for such preventive strategies in countries with limited resources becomes difficult.

### 14.15. Post-Resuscitation Immune Dysregulation

One of the emerging concepts that is being actively researched currently is the post-resuscitation immune dysregulation. Patients successfully resuscitated from cardiac arrest develop a myriad of features such as brain injury, myocardial dyfunction and systemic ischemic-reperfusion injury that occur as a result of the alteration of the immune system [122]. Studies have demonstrated high levels of circulating cytokines and adhesion molecules, and plasma endotoxin that lead to this immune dysregulation [123]. Further understanding of this dysregulation might help address the numerous complications that arise secondary to it such as VAP.

## 15. Limitations

One of the significant limitations we noted was that despite the plethora of evidence analyzing the association of VAP in Cardiac arrest, most of the literature pointed towards low certainty due to the lack of a uniform definition for VAP, thereby posing a challenge to establishing diagnosis. There also needed to be more uniformity in the treatment protocols due to the varying antibiotics used, the strength of antibiotics, and the duration of treatment. In addition, this review was limited to articles in the English language. Some of the recommendations are based on expert reviews and editorials that may be subject to bias (Appendix A).

## 16. Conclusions

This narrative review provides an update on the epidemiology, pathogenesis, diagnostic criteria, management, and VAP prevention strategies in patients suffering from cardiac arrest. Although, over the years, we have seen a significant reduction in the incidence of VAP in mechanically ventilated patients, among the patients who suffer from cardiac arrest, they still significantly contribute to the mortality. The increasing threat of muti-drug-resistant organisms raises further concerns in managing VAP. There is a need for standardized definitions of VAP to increase generalizability of studies. This could pave the way for Multi-center randomized trails which would help provide strong evidence regarding the difference diagnostic and management strategies employed. A global collaboration would further help address the potential antibiotic resistance and resource disparities which would in turn help curb the healthcare burden associated with VAP in cardiac arrest patients across the globe.

## 17. Future Direction

There has been an increased influx of research into the management and prevention strategies of VAP in cardiac arrest patients. The use of pathogen-specific antibodies for multi-drug resistant organisms have been upcoming. Animal studies have shown promise in the utility of nebulized bacteriophages in reducing lung bacterial burdens and also the treatment of methicillin-resistant bacteria. Studies to understand the pathogenesis of VAP in cardiac arrest patients are quintessential in formulating further management strategies. Well-designed randomized control trials will help validate existing strategies and treatment protocols.

## Figures and Tables

**Figure 1 medicina-61-00078-f001:**
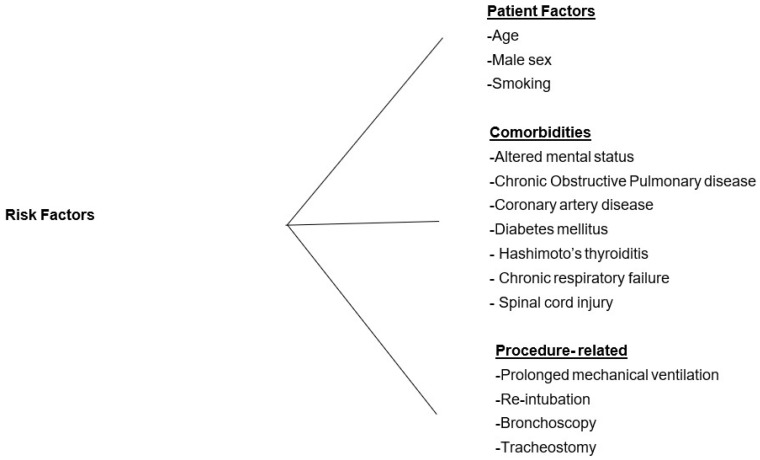
Risk Factors for VAP.

**Figure 2 medicina-61-00078-f002:**
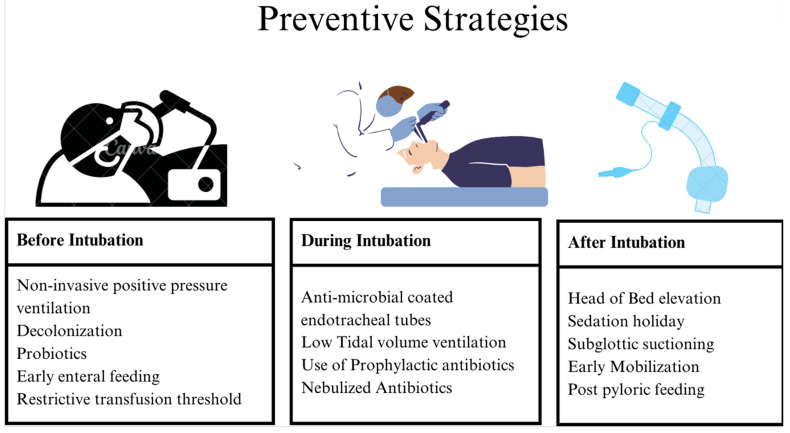
Preventive strategies for reducing the risk of VAP.

**Table 1 medicina-61-00078-t001:** Definition of pneumonia.

Term		Definition
CAP		An acute pulmonary infection acquired before hospitalization or within 48 h of hospitalizaiton
HAP	VAP	An acute pulmonary parenchymal infection occurred ≥ 48 h after endotracheal intubation
	nvHAP	An acute pulmonary infection in hospitalized patients after 48 h of hospitalization without mechanical ventilation.
HCAP		Retired term, used to classify pneumonia in patients with recent exposure to health care facilities including nursing homes and hemodialysis centers.

CAP: community-acquired pneumonia; HAP: Hospital-acquired pneumonia; VAP: Ventilator-associated pneumonia; nvHAP: non-ventilator hospital-acquired pneumonia; HCAP: Health care-associated pneumonia.

**Table 2 medicina-61-00078-t002:** Etiologic pathogens in VAP.

Polymicrobial	10–63.9%
*Monomicrobial isolate*	
*Gram-positive cocci*	
*Staphylococcus aureus*	9–44%
MRSA	4–25%
MSSA	3–17%
*Streptococcus pneumoniae*	1–12%
*Enterococcus faecalis*	0.8–5%
*Gram-negative bacilli*	
*Pseudomonas aeruginosa*	10–55%
*Acineteobacter* spp *	4–55%
*Haemophilus influenzae* ^¶^	8–26%
*Escherichia coli*	2–13%
Other Enterobacteriaceae	1–7%
Fungi	
*Candida* spp.	2–7%
*Aspergillus* spp.	2%

* Middle East, India, South America, and East Asia have higher prevalence; ^¶^ Haemophilus was found more prominent in early onset VAP.

**Table 3 medicina-61-00078-t003:** Differences between the 2017 International ERS/ESICM/ESCMID/ALAT guidelines and the 2016 ATS guidelines.

2016 Clinical Practice Guidelines by the Infectious Diseases Society of America and the American Thoracic Society [18]	2017 International ERS/ESICM/ESCMID/ALAT Guidelines [16]
(1) Suggests noninvasive sampling before initiating empiric antibiotic therapy with semiquantitative cultures to diagnose VAP, rather than invasive sampling with quantitative cultures and noninvasive sampling with quantitative cultures (weak recommendation, low-quality evidence). (2) For patients with suspected VAP whose invasive quantitative culture results are below the diagnostic threshold for VAP, we suggest that antibiotics be withheld rather than continued (weak recommendation, very low-quality evidence)	(1) Recommends obtaining a lower respiratory tract sample (distal quantitative or proximal quantitative or qualitative culture), to focus and narrow the initial empiric antibiotic therapy. (strong recommendation, low quality of evidence). (2) Suggests obtaining distal quantitative samples (prior to any antibiotic treatment) in order to reduce antibiotic exposure in stable patients with suspected VAP and to improve the accuracy of the results. (Weak recommendation, low quality of evidence).
(3) There is no evidence that invasive microbiological sampling with quantitative cultures improves clinical outcomes compared with noninvasive sampling with either quantitative or semiquantitative cultures. Noninvasive sampling can be done more rapidly than invasive sampling, with fewer complications and resources. Semiquantitative cultures can be done more rapidly than quantitative cultures, with fewer laboratory resources and less expertise needed. For these reasons, noninvasive sampling with semiquantitative cultures is the microbiological sampling technique recommended by the panel.	(3) The panel placed greater value on the potential benefits of reducing antibiotic exposure (and its impact on antibiotic resistance) than on the potential complications of invasive techniques.

**Table 4 medicina-61-00078-t004:** Comparing the advantages and disadvantages of clinical and microbiological strategies for the diagnosis of VAP.

	Advantages	Disadvantages
Clinical strategy	Decreases the risk of not treating patients with pneumonia (increased sensitivity)	Since the clinical features and imaging findings have low specificity, this approach leads to more antibiotic use, thus increasing the risk of resistant pathogens and recurrent pneumonia.
Microbiological strategy	Reduced use of antibiotics, thus decreasing the emergence of resistant pathogens, and avoiding the side effects of antibiotic use.	False-negative results can occur after recent antibiotic use in the past 24 h, but up to 72 h and in early forms of infection can lead to undertreatment of pneumonia. Delays in obtaining culture data can lead to complications. Studies have shown inconsistent results on repeating even invasive sampling methods If a noninvasive sampling method or a semiquantitative culture is used, it can lead to overtreatment since an infection cannot be differentiated from a colonization.

**Table 5 medicina-61-00078-t005:** Comparison of 2016 Clinical Practice Guidelines by IDSA/ATS and the the 2017 International ERS/ESICM/ESCMID/ALAT guidelines on empiric antibiotic therapy.

2016 Clinical Practice Guidelines by the Infectious Diseases Society of America and the American Thoracic Society [18]	2017 International ERS/ESICM/ESCMID/ALAT Guidelines [16]
Recommends empiric therapy to cover *Staphylococcus aureus* (MSSA), *Pseudomonas aeruginosa*, and other gram-negative bacilli with one of the following agents, piperacillin-tazobactam, cefepime, levofloxacin, imipenem or meropenem (strong recommendation, low-quality evidence)	Suggests narrow-spectrum empiric antibiotic therapy like ceftriaxone, cefotaxime, moxifloxacin or levofloxacin if there is a low risk for MDR pathogens and if it is early-onset VAP (weak recommendation, very low-quality evidence)
Recommends adding vancomycin or linezolid to cover for MRSA (strong recommendation, moderate quality evidence) * if - there is a risk factor for antimicrobial resistance(Table 2) - it is a unit where >10–20% of S.aureus are methicillin-resistant - it is a unit where the prevalence of MRSA is unknown (weak recommendation, very low-quality evidence)	Recommends combination therapy to cover MRSA and *P. aeruginosa* if the patient has septic shock and/or the following risk factors for potentially resistant microorganisms - hospital settings with high rates of MDR pathogens (i.e., a pathogen not susceptible to at least one agent from three or more classes of antibiotics) - previous antibiotic use, - recent prolonged hospital stay (>5 days of hospitalization) - previous colonization with MDR pathogens - Prevalence of >25% resistant pathogens in local microbiological data (strong recommendation, moderate quality evidence)
Suggests empiric therapy with two agents from different classes to cover for *Pseudomonas aeruginosa* ^, instead of one, if - there is a risk factor for antimicrobial resistance (Table 2) - it is a unit where >10% of gram- negative organisms are resistant to an agent being considered for monotherapy - local antimicrobial susceptibility rates are not available - if patient has structural lung disease increasing the risk of gram negative bacilli infection, like bronchiectasis or cystic fibrosis (weak recommendation, low quality evidence)	
Avoid aminoglycoside for *P. aeruginosa* (strong recommendation, very low quality evidence) and any gram-negative bacilli coverage if alternative agents are available due to the poorer clinical response rates and higher risk of nephrotoxicity (weak recommendation, low-quality evidence)	
Avoid colistin if alternative agents are available for gram-negative bacilli are available since overuse can jeopardize its current role as the last resort and has increased risk of nephrotoxcity (weak recommendation, very low quality evidence).	

* Vancomycin and linezolid showed similar outcomes, and the choice depends on patient-specific factors [60,61,62]; ^ Antipseudomonal cephalosporin (cefepime, ceftazidime) or Antipseudomonal carbapenem (imipenem or meropenem) or Beta lactam/Beta lactam inhibitors (piperacillin-tazobactam) PLUS Antipseudomonal fluoroquinolone (ciprofloxacin or levofloxacin) or Aminoglycoside (amikacin, gentamicin, or tobramycin) [15].

## Data Availability

No new data were created or analyzed in this study.

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
