# Peer review of "Ventilator-Associated Pneumonia After Cardiac Arrest and Prevention Strategies: A Narrative Review"

_medicina, 2025, doi:10.3390/medicina61010078_

Round 1
Reviewer 1 Report
Comments and Suggestions for Authors
A well written literature synthesis of the available data around the subject; I think that the present manuscript could be published in its present form.
Author Response
A well written literature synthesis of the available data around the subject; I think that the present manuscript could be published in its present form.
Thank you very much for your comment.
Reviewer 2 Report
Comments and Suggestions for Authors
This submission aimed to comprehensively review VAP after cardiac arrest. It is a good review. However, I had several suggesitons.
Major
1. I think many of the cited study focussed on VAP, rather than VAP after cardiac arrests. Please confirm all the information was based on the patients after cardiac arrests, particulary for the epidemiology section.
Minor
1. All bacteial name should be presented in italic
2. Please add the role of molecular diagnosis of microbiological investigaitons for patients with VAP
3. Please discuss the role of using VAE to define VAP.
4. In the preventive section, please add more detailed data.
Author Response
- I think many of the cited study focussed on VAP, rather than VAP after cardiac arrests. Please confirm all the information was based on the patients after cardiac arrests, particulary for the epidemiology section.
Thank you for the astute comment. Although we have mentioned few studies that are based on general VAP patients, they have been mentioned in the epidemiology section in order to elucidate general information on the morbidity and mortality on VAP and then address the VAP incidence in cardiac arrest patients specifically.
(Page 3, Line 112: We have removed certain lines that were based on general VAP patients
Page 4, Line 122: We have removed certain lines that were based on general VAP patients)
- All bacteial name should be presented in italic
Thank you for the comment. We have made the revision in all the areas of the manuscript as below
(Page 2, Line 82: “Streptococcus pneumoniae takes precedence as the most prevalent bacterial cause, closely followed by Haemophilus influenza and Moraxella catarrhalis.”
Page 3, Line 93: “Pathogens are similar among VAP and nvHAP, with the most common organisms being Staphylococcus aureus (including methicillin-resistant S. Aureus) and Pseudomonas aeruginosa.”
Page 4 , Line 146: “Despite this diversity, a consistent pattern emerges across numerous studies, with Pseudomonas aeruginosa (10-55%) and Staphylococcus aureus (7-44%) being the predominant pathogens implicated in VAP cases and less commonly by streptococcus pneumoniae, enterococcus faecalis, and fungal.”
Page 4, Line 166: While the prevalence of Pseudomonas aeruginosa in VAP remains relatively consistent worldwide, certain regions, including the Middle East, India, East Asia, and South America, exhibit a distinctive prevalence of Acinetobacter baumannii as a prominent infectious etiology in VAP
Page 5, Line 173: Staphylococcus aureus is commonly present in the orolaryngeal and tracheobronchial flora.
Page 4, Line 183: This, in turn, significantly increases the likelihood of carbapenem-resistant infections, particularly involving Pseudomonas aeruginosa (21%), Klebsiella pneumonia (14%), and Acinetobacter baumannii (22.5%).
Page 11, Line 375: Combination regimens have been used for reasons like achieving synergy in the therapy of P. Aeruginosa and preventing the emergence of resistance during therapy.
Page 11, Line 393: However, for P. Aeruginosa and Acinetobacter, the relapse rate was higher with a shorter duration of treatment.
Page 12, Line 404: If P. Aeruginosa is suspected and there is a risk for high-risk disease, combination therapy including drugs from two different classes is recommended by the 2016 Clinical practice guidelines by IDSA/ATS)
- Please add the role of molecular diagnosis of microbiological investigaitons for patients with VAP
Thank you for the excellent comment and we have added the following statement.
(Page 8, Line 299: One of the major drawbacks of the microbiological culture assays was the yield and the delay in the time of diagnosis. Hence, molecular diagnostic methodologies such as Polymerase chain reaction (PCR) can be utilized to rapidly diagnose patients and ascertain the diagnosis.)
- Please discuss the role of using VAE to define VAP.
Thank you for the astute comment. We have incorporated the following statement.
(Page 3, Line 99: “The United States centers for Disease Control and Prevention (CDC) created the ventilator associated events (VAE) definition to replace the traditional definitions for ventilatory associated pneumonia (VAP). Although a number of studies have been performed to assess the overlap between VAE and VAP, there appears to be a mismatch. Hence, VAE has been defined to be a new entity as a safety surveillance tool to encompass a broad range of potential nosocomial complications. )
- In the preventive section, please add more detailed data.
We agree with the comment and we have added more information to the preventive section.
(Page 14, Line 502 : “ Nava et al demonstrated that noninvasive pressure support during weaning decreases the incidence of pneumonia and also improve 60 day mortality. Antonelli et demonstrated that patients on noninvasive ventilation improved gas exchange as well as had fewer complications and shorter ICU stays. ”.
Page 14, Line 509: “ Supine positioning in addition to long duration of mechanical ventilation and decreased consciousness lead to increased risk of aspirations. Drakulovic et al. first demonstrated that semirecumbent body position reduced the frequency and risk of VAP.93 This study was based on the fact that several studies showed increased aspiration risk in supine patients. However semirecumbent positioning does not protect completely from gastroesophageal reflux. ”.
Page 15, Line 525 : “ The Study by Brook et al showed that protocol directed sedation reduced the duration of mechanical ventilation as well as ICU and hospital stay. They attributed it to reduction in complications such as VAP incidence. ”.
Page 16, Line 563 : “ Their study showed that patients treated with Lactobacillus rhamnosus were less likely to to develop microbiologically confirmed VAP”
Page 16, Line 570 : “ The study by Wang et al showed that early mobilization reduced the incidence of intensive care unit-acquired weakness and complications such as VAP, deep vein thrombosis, pressure sores and also reduced the duration of mechanical ventilation, length of ICU stay and hospital stay ”.
Page 16, Line 578 : “ Special attention was recommended to the provision of glutamine and omega-3 fatty acids as well as the care of patients such as those with dysphagia, frail patients, multiple trauma patients, abdominal surgery, sepsis and obesity. ”.
Page 16, Line 584 : “ 41 studies were included across 10 countries involving 3248 patients and the results showed that post-pyloric feeding was associated with lower rate of pulmonary aspiration, gastric reflux, gastrointestinal complications and VAP ”.
Page 16, Line 591 : “ Neto et al studied patients on low tidal volume ventilation compared to those not on it and found a lower number of ICU free days, hospital free days and a greater mortality and complication rate amongst patients not on low tidal volume ventilation. ”.)
Reviewer 3 Report
Comments and Suggestions for Authors
Dear authors, thanks for the opportunity to revise your narrative review entitled: "Ventilator-Associated Pneumonia after Cardiac Arrest and Prevention Strategies: A Narrative Review". Authors have done great job writing this narrative review on existing literature on the topic. Anyway, there are some concerns that should be addressed before reconsideration for publication:
- Please revise English language through out the manuscript.
- Please clarify better where where the evidence is strongest and where gaps remain. Try to use a hierarchy of evidence to distinguish high-quality findings from weaker recommendations.
- Please, consolidate overlapping topics into unified sections. For example, combine subglottic suctioning and antimicrobial-coated tubes under a broader category like "Endotracheal Tube Management." Similarly, group all aspiration prevention strategies (e.g., head-of-bed elevation, post-pyloric feeding) together.
- While the review acknowledges antibiotic resistance as a concern (e.g., with prophylactic antibiotics and SDD), this issue is not adequately explored. The long-term implications of resistance on VAP prevention strategies are under-discussed.
- The review does not address variations in VAP incidence, management, or prevention across different healthcare systems, particularly in low-resource settings. Please, provide a more global outlook by discussing how resource constraints may impact the feasibility of interventions like antimicrobial-coated tubes or advanced diagnostic tools.
-Please discuss unique risk factors for cardiac arrest patients, such as post-resuscitation immune dysregulation.
- Please, address challenges in diagnosing VAP in cardiac arrest patients, who may have overlapping symptoms with other conditions.
- Conclusion and future direction: the conclusion could explicitly call for:
- Standardized definitions of VAP for uniform research.
- Multi-center randomized trials specific to cardiac arrest patients.
- Global collaboration to address antibiotic resistance and resource disparities.
Author Response
- Please revise English language through out the manuscript.
Thank you for your astute comment. We have revised the entire manuscript and revised the English language accordingly
- Please clarify better where where the evidence is strongest and where gaps remain. Try to use a hierarchy of evidence to distinguish high-quality findings from weaker recommendations.
Since this is a narrative review and not a systematic review, the authors have not conducted a formal synthesis and grading of certainty in evidence based on the GRADE guidelines. However, based on the reviewer’s request we have added a supplementary table highlighting studies (randomized trials) with high to moderate degree of certainty compared to studies (observational retrospective) with moderate to low degree of certainty.
|
High to moderate degree of certainty |
Moderate to low degree of certainty. |
|
|
- Please, consolidate overlapping topics into unified sections. For example, combine subglottic suctioning and antimicrobial-coated tubes under a broader category like "Endotracheal Tube Management." Similarly, group all aspiration prevention strategies (e.g., head-of-bed elevation, post-pyloric feeding) together.
Thank you for your keen observation. All the topics that you have mentioned are the different strategies that can employed to prevent aspiration. In order to clarify this better, we have included a new heading as Aspiration prevention strategies.
Page 14, Line :498 ( Aspiration Prevention Strategies)
- While the review acknowledges antibiotic resistance as a concern (e.g., with prophylactic antibiotics and SDD), this issue is not adequately explored. The long-term implications of resistance on VAP prevention strategies are under-discussed.
We agree with the comments of the reviewer. We have added the following statement to provide more detail on antibiotic resistance.
(Page 17, Line 643 “Several of the VAP prevention strategies encompass the utilization of antibiotics in systemic and topical modalities to prevent the incidence of VAP. Although several of these strategies have been proven to be beneficial, there is always greater concerns of antibiotic resistance after long term use of antibiotic resistance. We would require studies of longer duration to assess the impact of these strategies in the long term antibiotic stewardship. “.)
- The review does not address variations in VAP incidence, management, or prevention across different healthcare systems, particularly in low-resource settings. Please, provide a more global outlook by discussing how resource constraints may impact the feasibility of interventions like antimicrobial-coated tubes or advanced diagnostic tools.
Thank you for your comment. The Manuscript has been revised to include the following statement.
(Page 9,Line 321:” Although there has been a recent increase in the utilization of different diagnostic strategies and resources to identify VAP earlier and in turn help reduce the significant healthcare burden associated with increased hospital and ICU stay, several of these strategies have a high cost associated with them. This reduces the availability and accessibility of these resource in developing countries and resource limited economies. ”.)
(Page 18,Line 650: The other limiting factor that is associated with the utilization of these management strategies is availability of health care funding to invest in one or several of these modalities. The health expenditures per capita vary largely between developing and developed countries and hence allocation of funds for such preventive strategies in countries with limited resources becomes difficult. )
6.Please discuss unique risk factors for cardiac arrest patients, such as post-resuscitation immune dysregulation.
Thank you for the excellent suggestion. We have revised the manuscript to add the following discussion about post-resuscitation immune dysregulation.
(Page 18, Line 656: “One of the emerging concepts that is being actively researched currently is the post-resuscitation immune dysregulation. Patients successfully resuscitated from cardiac arrest develop a myriad of features such as brain injury, myocardial dyfunction and systemic ischemic-reperfusion injury that occur as a result of the alteration of the immune system.130 Studies have demonstrated high levels of circulating cytokines and adhesion molecules, and plasma endotoxin that lead to this immune dysregulation. 131 Further understanding of this dysregulation might help address the numerous complications that arise secondary to it such as VAP. ”.)
- Please, address challenges in diagnosing VAP in cardiac arrest patients, who may have overlapping symptoms with other conditions.
Thank you for your comment. The Manuscript has been revised to include the following statement.
(Page 4,Line 133:” Diagnosis of VAP in patients with cardiac arrest is complex due to the nonspecific nature of its clinical presentation, which often overlaps with other conditions commonly seen in critically ill patients. Fever, leukocytosis, hypoxemia, and radiographic infiltrates, hallmarks of VAP, can also be caused by noninfectious processes such as acute respiratory distress syndrome (ARDS), atelectasis, pulmonary thromboembolism, pulmonary hemorrhage, or drug-induced pulmonary injury. Similarly, systemic conditions like congestive heart failure or hemodynamic instability following cardiac arrest can mimic VAP symptoms, further complicating diagnosis ”.)
8.Conclusion and future direction: the conclusion could explicitly call for:
- Standardized definitions of VAP for uniform research.
- Multi-center randomized trials specific to cardiac arrest patients.
- Global collaboration to address antibiotic resistance and resource disparities.
The conclusion has been revised to incorporate the above mentioned strategies.
(Page 18, Line : 681, “ This could pave the way for Multi-center randomized trails which would help provide strong evidence regarding the difference diagnostic and management strategies employed. A global collaboration would further help address the potential antibiotic resistance and resource disparities which would in turn help curb the healthcare burden associated with VAP in cardiac arrest patients across the globe. “ )
Round 2
Reviewer 2 Report
Comments and Suggestions for Authors
The authors response well.
Reviewer 3 Report
Comments and Suggestions for Authors
Authors have addressed all the comments raised.
Thanks again for the opportunity to revise this paper.